# Longitudinal symptom profile of palliative care patients receiving a nurse-led end-of-life (PEACH) programme to support preference to die at home

Meera Agar,[1] Wei Xuan,[2] Jessica Lee,[3] Gregory Barclay,[4] Alan Oloffs,[5] Kim Jobburn,[1] Janeane Harlum,[1] Nutan Maurya,[1] Josephine Sau Fan Chow [6]

For numbered affiliations see end of article.

**Correspondence to**
Dr Josephine Sau Fan Chow;
josephine.chow@health.nsw.gov.au

## ABSTRACT

**Objectives** Tailored models of home-based palliative care aimed to support death at home, should also ensure optimal symptom control. This study aimed to explore symptom occurrence and distress over time in Palliative Extended And Care at Home (PEACH) model of care recipients.

**Design** This was a prospective cohort study.

**Setting and participants** Participants were consecutive recipients of the PEACH rapid response nurse-led model of care in metropolitan Sydney (December 2013–January 2017) who were in the last weeks of life with a terminal or deteriorating phase of illness and had a preference to be cared or die at home.

**Outcome measures** Deidentified data including sociodemographic and clinical characteristics, and symptom distress scores (Symptom Assessment Score) were collected at each clinical visit. Descriptive statistics and forward selection logistic regression analysis were used to explore influence of symptom distress levels on mode of separation ((1) died at home while still receiving a PEACH package, (2) admitted to a hospital or an inpatient palliative care unit or (3) discharged from the package (alive and no longer requiring PEACH)) across four symptom distress level categories.

**Results** 1754 consecutive clients received a PEACH package (mean age 70 years, 55% male). 75.7% (n=1327) had a home death, 13.5% (n=237) were admitted and 10.8% (n=190) were still alive and residing at home when the package ceased. Mean symptom distress scores improved from baseline to final scores in the three groups (p<0.0001). The frequency of no symptom distress score (0) category was higher in the home death group. Higher scores for nausea, fatigue, insomnia and bowel problems were independent predictors of who was admitted.

**Conclusion** Tailored home-based palliative care models to meet preference to die at home, achieve this while maintaining symptom control. A focus on particular symptoms may further optimise these models of care.

## BACKGROUND

Symptom burden is one of the most distressing aspects of the end-of-life (EOL) experience for people with a palliative diagnosis and for their families and loved ones.[1 2] The symptoms which are prevalent near the EOL are well documented and include breathlessness and respiratory secretions, pain, fatigue, anorexia and gastrointestinal symptoms.[2 3]

Home-based palliative care services have been expanding worldwide, and many people at the EOL prefer to be cared for and in many cases also die at home.[4–6] Community palliative care services have been demonstrated to improve patients and caregivers' experience at EOL through better symptom control, coordination of care and improved communication between professionals, patient and family and increase the odds of home deaths.[7 8] A recent review also suggests the strongest evidence for cost effectiveness relates to home-based interventions through both decreased healthcare costs and resource use and improvements in patient and caregiver outcomes[9] However, they also indicated that there is still room to improve home palliative care services and increase benefits to patients, inclusive of those who wish to die at home.[8–10] A study of 25 679 patients who died under the care of a hospital or home-based

**STRENGTHS AND LIMITATIONS OF THIS STUDY**

⇒ This study is one of the largest cohorts of a tailored model of care aimed at improving the rates of home death for those where it was their expressed preference.
⇒ The symptom assessments were collected as part of routine clinical care using a reliable and valid measure designed to be integrated into routine clinical care.
⇒ Only symptoms assessed by the Symptom Assessment Scale were included in the analysis.

palliative care team in Australia found that though over 85% of all patients had no severe symptoms prior to death, patients at home had less improvement over all and some symptoms got worse.[11]

The *Palliative Extended And Care at Home* (PEACH) model of care (a rapid response nursing care package delivered in conjunction with existing palliative care community services) was developed for those who had high or complex palliative care needs to support care at home. Evaluation demonstrated it achieved increased days at home compared with usual care, with the PEACH model of care costs offset by reduced costs due to lower inpatient care needs.[12 13] The PEACH model of care was then further tailored to provide EOL care in the last week of life for those patients who had an expressed wish to die at home. A prospective cohort study found that the majority of PEACH recipients with a clear preference to die at home when the service was initiated were able to die at home.[14] Though preference for place of death was achieved, it is also important to understand if symptom control was optimal for those who received this model of care.

This study aimed to evaluate symptom occurrence and level of symptom distress over time in recipients of the PEACH model of care, and explore the associations of symptom distress with mode of separation.

## METHODS
### Study design
Prospective cohort study.

### Study population
Patients registered with the specialist palliative care services in five Local Health Districts (LHDs) in metropolitan Sydney were eligible to receive a PEACH package if they were assessed as likely to be in the last week of life (terminal or deteriorating phase of illness[15] and Australia-modified Karnofsky Performance Scale score≤40,[16] have burdensome symptoms and/or require increased level of support, and had a preference to die at home. Consecutive recipients of the PEACH package between December 2013 and January 2017 were included.

### Patient and public involvement
Patients or the public were not involved in the design, or conduct, or reporting, or dissemination plans of our research.

### Study setting
The PEACH Program is coordinated through one LHD in metropolitan Sydney, Australia to provide packages across five New South Wales metropolitan LHDs (South Western Sydney, Illawarra Shoalhaven, Nepean Blue Mountains, Sydney and Western Sydney) with PEACH nursing services delivered in partnership with a non-government organisation (Silver Chain Group). Australia

provides universal healthcare (Medicare), which may be supplemented by private health insurance.

### Study intervention
The PEACH Program commenced in 2013 funded by the New South Wales Government, Australia in the participating health districts to address the needs of the community with a palliative diagnosis to facilitate a chosen home death while maintaining quality in the EOL care.[14] PEACH enables clients in their deteriorating and/or terminal phase to die at home or to stay at home as long as possible as per their wishes through provision of intensive, rapid, flexible, individualised nurse-led care and coordination for EOL care (last 7 days or can be extended) while maintaining satisfactory symptoms control and function at home.

Patients who were undecided about their preferred place of death but expressed an interest in attempting to stay home for EOL care or who wished to remain at home and transfer to hospital when death was imminent were also provided with a PEACH package. It was recognised for some clients there are a number of factors such as confidence in ability to die at home by either the patient and/or carer as well as cultural, religious, education or family demographics such as younger children in the home, impacted on preference for EOL care, that is, undecided or transfer when death was imminent. Achieving EOL care preference was considered as one of the main objectives of the programme.

The PEACH package is centrally coordinated through a hub which coordinates community care services including community palliative care. The package provides some services in addition to existing services delivered by general practitioners, community nurses and specialist palliative care teams. The additional services were provided by a PEACH assistant in nursing during day time to provide either personal care or some respite to the carer for up to 1 hour/day, 7 days/week and a PEACH registered nurse visit in the evening and overnight telephone or video support by an experienced palliative care nurse who were fully briefed of the patient's care plan until the client separated from the programme.

### Data collection and variables
Data were collected routinely for all consecutive PEACH package recipients until separation from the package. Deidentified data were used from administrative and clinical information systems obtaining the prospectively collected standardised data requirements and clinical assessments which were a requirement at referral to the PEACH programme and during delivery of the PEACH package. To ensure confidentiality and anonymity, data were collected using an alpha-numeric code for each participant.

### Symptom occurrence and symptom distress
All patients as per the model of care were in receipt of daily clinical visits, which included a daily symptom score

assessment until separation from the programme. At times, some clients/carers declined the daily visit and hence Symptom Assessment Scale (SAS) would not be recorded for that day.

Symptom distress was assessed using SAS, which measures the perceived distress from common symptoms experienced by people receiving palliative care.[17] It is a valid and reliable patient reported outcome measure suitable for routine clinical care with patients requiring palliative and EOL[17 18] in Australia. The scale assessed eight symptom dimensions: pain, insomnia (difficulty sleeping), nausea, vomiting, bowel problems, appetite problems, breathing problems and fatigue on a 11 point numerical scale with zero being no distress due to that symptom and 10 being the worst distress possible. Symptom distress was categorised into four levels based on the score for each symptom dimension (0, >0 to ≤3, >3 to ≤6 and >6 to ≤10).

### Mode of separation

Mode of separation was defined as the status of the patient at the end of the PEACH package episode of care (discharge, transfer or death) and the place to which the person is transferred.[19] Three possible modes of separation were identified from the PEACH package, namely (1) person died at home while still receiving a PEACH package, (2) was admitted to a hospital or inpatient palliative care unit (PCU) or if (3) were discharged from the package (alive and no longer requiring PEACH).

### Sociodemographic and clinical variables

At the time of referral to the PEACH programme, sociodemographic clinical data were collected including age, gender, caregiver relationship to client (spouse, child, grandchild, other relative, others), preference for place of death on referral (classified as home, hospital/PCU, or undecided), diagnosis (malignant, non-malignant), palliative care phase on referral, location of referral (hospital/PCU, community), language spoken (English, Some English, no English), and as a proxy measure of geographical distance to admitted hospital services local government area (metropolitan, inner and outer regional, remote or very remote as defined by Australian Classification of Local Governments determined by the Australian Bureau of Statistics).

### Statistical analyses

Data was analysed using SAS software (V.9.4). Clinical and demographic characteristics were summarised using frequency counts and percentages for categorical data and the mean and SD for continuous variables.

Mean SAS Scores at baseline (on referral) and final (average of the two scores recorded prior to separation) were calculated and compared for each symptom by the frequency and four levels of symptom distress by mode of separation (PEACH recipients who died at home vs those who were admitted to hospital or PCU as mode of separation). Forward selection logistic regression analysis were used to explore influence of symptom distress levels on mode of separation across the four symptom distress-level categories.

## RESULTS

### Recipients of the PEACH model of care

A total of 1754 consecutive palliative care patients (table 1) received a PEACH package across the 5 LHDs from December 2013 to January 2017.

The full participant characteristics are described elsewhere[14] but in summary the majority had a cancer diagnosis, mean age was 70 years (0–106), and 55.4% (n=978) of them were male. Most patients reported their preferred place of death to be home (89%, n=1561) with smaller numbers citing the hospital or PCU (3.8%, n=67), and 7.2% (n=126) of the clients were not sure about their preferences. Overall, 75.7% (n=1327) clients died at home, 13.5% (n=237) were admitted to hospital or PCU and 10.8% (n=190) were alive at the end of the episode of care from PEACH programme.

Almost, 78.7% of the clients who indicated they preferred to die at home met their wish; 55.6% of clients who were unsure/undecided of preferred place of death on referral to the programme died at home (table 1).

The duration of stay on the programme for clients ranged from <24 hours to 2 months. The median duration on the programme was 6 days. The duration of the stay was same irrespective of the mode of separation.

Patients who were discharged and came back onto the programme at a later date, they were counted as another patient encounter. Thirty-three clients who were discharged from the programme rejoined later.

### Symptom distress over time by mode of separation

In the group who died at home, the mean symptom distress scores all improved remaining in the mild symptom distress category, though fatigue showed the least improvement and was rated as moderate levels of symptom distress (table 2). A similar pattern was seen for those who were admitted to a hospital or PCU, or who were discharged from the PEACH package except for fatigue which worsened in both groups.

When the symptom distress scores were compared for the three modes of separation, no significant differences in the baseline score was observed by mode of separation. However, a significant differences in the symptom distress scores prior to separation (final score) was observed for all symptoms by mode of separation (table 2).

The final symptom distress scores by the four categories (score=0, >0 to ≤3, >3 to ≤6 and >6 to ≤10) for each symptom for the two modes of separation (died at home, admitted) is outlined in table 3. The other three categories (>0 to ≤3, >3 to ≤6 and >6 to ≤10) were compared with the=0 category. The frequency of no symptom distress score (0) category was higher in the group who died at home, as compared with the group who were admitted to hospital. The higher score category for nausea, fatigue,

**Table 1** Sociodemographic characteristics at referral to the PEACH programme by the three modes of separation

| | Died at home (n=1327) | Admitted to hospital or palliative care unit (n=237) | Alive and no longer requiring PEACH (n=190) |
|---|---|---|---|
| Age, mean (SD) | 70.8 (14.7) | 69.3 (14.2) | 68.4 (18.0) |
| Gender (n, %) | | | |
| Female | 605 (45.6) | 96 (40.5) | 83 (43.7) |
| Male | 722 (54.4) | 141 (59.5) | 107 (56.3) |
| Diagnostic category (n, %) | | | |
| Cancer | 1094 (82.5) | 214 (90.3) | 146 (76.8) |
| Non-cancer | 232 (17.50) | 23 (9.7) | 44 (23.2) |
| Referral source (n, %) | | | |
| Community Health Centre | 1014 (76.4) | 174 (73.4) | 130 (68.4) |
| Hospital/Palliative Care Unit—Ward/Others | 313 (23.6) | 63 (26.6) | 60 (31.6) |
| Language (n, %) | | | |
| English | 1126 (84.9) | 207 (87.3) | 160 (84.2) |
| Some English | 77 (5.8) | 13 (5.5) | 8 (4.2) |
| Non-English speaking | 124 (9.3) | 17 (7.2) | 22 (11.6) |
| Preferred place of death (n, %) | | | |
| Home | 1229 (92.6) | 181 (76.4) | 151 (79.5) |
| Hospital/Palliative Care Unit | 28 (2.1) | 25 (10.6) | 14 (7.4) |
| Undecided | 70 (5.3) | 31 (13.1) | 25 (13.2) |
| LGA category (n, %) | | | |
| Major city | 1013 (76.3) | 195 (82.3) | 154 (81.1) |
| Inner regional | 288 (21.7) | 39 (16.5) | 35 (18.4) |
| Outer regional | 26 (2.0) | 3 (1.3) | 1 (0.5) |
| Carer relationship (n, %) | | | |
| Spouse | 657 (50.0) | 150 (63.8) | 97 (51.6) |
| Child/grand child | 517 (39.4) | 67 (28.5) | 70 (37.2) |
| Others | 96 (7.3) | 11 (4.7) | 16 (8.5) |
| Sibling | 43 (3.3) | 7 (3.0) | 5 (2.7) |

PEACH, Palliative Extended And Care at Home.

insomnia and bowel problems was associated with being admitted to hospital.

### Association between symptom distress and mode of separation

On univariate analyses the higher the final symptom distress score (average of last two scores recorded prior to separation), the higher the odds of patient being admitted to hospital or PCU as mode of separation, except for breathing problems which showed the reverse association (table 4).

On multivariate analysis with forward selection, higher final symptom distress scores for nausea, fatigue, insomnia and bowel problems were associated with being admitted to hospital (table 5).

### DISCUSSION

This study has demonstrated that a tailored model of care to support palliative care patients who have a preference to die at home supports excellent symptom control with symptom distress of mild severity and improving over time regardless of mode of separation from the PEACH package, apart from fatigue. The frequency of a rating of no symptom distress was highest in the group who died at home. The higher symptom distress for nausea, fatigue, insomnia and bowel problems was associated with being admitted to hospital. Symptom distress due to breathing problems were less likely to be admitted to hospital possibly reflecting the breathing changes in the imminently dying person where moving location of care would be inappropriate.

These findings are in the context of our prior study[14] finding that the majority of PEACH package recipients who have a clear preference to die at home when the service is initiated were able to achieve their goal to die at home. People who were undecided at referral or had a preference for admission when death was imminent were more likely to be admitted to hospital or a PCU than die at home. People with a non-cancer diagnosis or who were cared for by their child or grandchild or friend/relative as caregiver were less likely to be admitted to hospital/PCU as a mode of separation.

We found that higher symptom distress score increases the likelihood of patient being admitted to the hospital or

**Table 2** Comparison of baseline and final symptom distress scores by mode of separation

| | Died at home | | Admitted to hospital of palliative care unit | | Alive and no longer requiring PEACH | | |
| | n=1327 | | n=237 | | n=190 | | |
| | Mean | SD | Mean | SD | Mean | SD | P value |
|---|---|---|---|---|---|---|---|
| Baseline score | | | | | | | |
| Insomnia | 1.47 | 2.12 | 1.64 | 2.26 | 1.33 | 1.84 | 0.322 |
| Appetite | 2.16 | 2.74 | 2.37 | 2.79 | 2.45 | 2.78 | 0.297 |
| Nausea | 1.41 | 2.50 | 1.45 | 2.35 | 1.47 | 2.39 | 0.933 |
| Bowel problems | 1.55 | 2.28 | 1.49 | 2.32 | 1.39 | 2.11 | 0.651 |
| Breathing problems | 2.06 | 2.56 | 1.96 | 2.53 | 2.19 | 2.64 | 0.666 |
| Fatigue | 5.12 | 3.26 | 4.76 | 3.08 | 4.92 | 3.09 | 0.271 |
| Pain score | 1.89 | 2.26 | 1.84 | 2.36 | 1.60 | 2.07 | 0.477 |
| Vomiting score | 0.30 | 1.11 | 0.44 | 1.36 | 0.43 | 1.11 | 0.311 |
| Final score | | | | | | | |
| Insomnia | 0.69 | 1.69 | 1.24 | 2.21 | 0.73 | 1.58 | <0.0001 |
| Appetite | 1.05 | 2.58 | 1.75 | 2.55 | 1.25 | 2.09 | 0.0006 |
| Nausea | 0.18 | 0.87 | 0.51 | 1.49 | 0.52 | 1.46 | <0.0001 |
| Bowel problems | 0.79 | 1.73 | 1.35 | 2.13 | 0.76 | 1.58 | <0.0001 |
| Breathing problems | 2.00 | 2.41 | 1.35 | 2.29 | 1.09 | 1.76 | <0.0001 |
| Fatigue | 4.98 | 4.03 | 6.06 | 2.49 | 5.36 | 2.66 | 0.0002 |
| Pain score | 1.59 | 2.11 | 1.91 | 2.54 | 1.15 | 1.99 | 0.0018 |
| Vomiting score | 0.12 | 0.73 | 0.30 | 1.34 | 0.19 | 0.86 | 0.0112 |

PEACH, Palliative Extended And Care at Home.

PCU. This mirrors the findings of other studies which have reported that patients were more likely to be hospitalised or die in the hospital if they had higher functional status or greater pain intensity (pain score≥2)[20] or when symptoms such as pain and dyspnoea were present.[21] A recent home-based palliative care cohort study of adult patients with cancer in Singapore showed that high symptom needs increased incidence rate ratios of acute healthcare utilisation.[22] The reason that the patients sought admission to the hospital or PCU could be due to concerns in managing symptoms at home which might require a level of supervision and physical assistance beyond caregivers' coping capacity, or to seek clinical assessment and adjustment to management, with these decisions may have been supported by the clinicians involved in their care in some cases.

Studies demonstrate a significant decrease in the level of severe symptom distress after start of palliative care.[11 23] Notably, Gill et al[23] found no change in the rates of fatigue ascertained from monthly interviews of 665 decedents of community living older people, and symptom that we also found caused moderate symptom distress increasing over time. Eagar et al evaluated data derived from the Australian Palliative Care Outcomes Collaborative and found for 25 679 patients who died under the care of a hospital or home-based palliative care team symptom distress improved and over 85% of patients had no severe symptom distress (scores 8–10) just prior to death.[11] Similarly, they found high rates for severe symptom distress for fatigue with this occurring in 14.6% of those in hospital and 10% of those at home just before death.

There is limited literature exploring differences in severe symptom outcomes for palliative care patients receiving hospital care compared with those receiving care at home. Some studies report that symptom outcomes are better for hospital patients and patients at home have less improvement overall and some symptoms get worse. A retrospective cohort study of 359 patients with advanced cancer reported that those with final place of care as home had the lowest pain and depression scores as compared with patients in the inpatient hospice group or the hospital group.[20] A study in China found enhanced intensity of home palliative care visits with two additional days per week and formulated standard operating procedures for dyspnoea could significantly reduce the rate of ED visits by 30.7% (p<0.05) due to dyspnoea during the last 6 months of life.[24] Nausea and well-being scores were also significantly worse in the inpatient hospice group, compared with the home group.[20] It is not clear if those who were hospitalised had access to specialist palliative care. In contrast, Eagar et al found that palliative care patients under the palliative care teams in the hospital were 3.7 times less likely than those at home to have no severely distressing symptoms before death.[11] This suggests that supplementing specialist palliative care with bespoke packages for the last week of life can ensure severe distress from symptoms is negated.

**Table 3** Final symptom distress by category for those who died at home and those who were admitted

| Symptom distress category (score range) | Frequency, N (%) | | |
|---|---|---|---|
| | Died at home | Admitted | P value |
| Insomnia | n=1321 | n=237 | |
| =0 | 942 (87.1) | 139 (12.9) | |
| >0 to ≤3 | 266 (79.2) | 70 (20.8) | 0.0006 |
| >3 to ≤6 | 96 (82.1) | 21 (17.9) | |
| >6 to ≤10 | 17 (70.8) | 7 (29.2) | |
| Appetite | n=1271 | n=235 | |
| =0 | 880 (89.6) | 102 (10.4) | |
| >0 to ≤3 | 224 (71.8) | 88 (28.2) | <0.0001 |
| >3 to ≤6 | 106 (76.3) | 33 (23.7) | |
| >6 to ≤10 | 61 (83.6) | 12 (16.4) | |
| Nausea | n=1318 | n=237 | |
| =0 | 1182 (86.8) | 180 (13.2) | |
| >0 to ≤3 | 112 (72.3) | 43 (27.7) | <0.0001 |
| >3 to ≤6 | 19 (65.5) | 10 (34.5) | |
| >6 to ≤10 | 5 (55.6) | 4 (44.4) | |
| Bowel problem | n=1322 | n=237 | |
| =0 | 836 (88.3) | 111 (11.7) | |
| >0 to ≤3 | 370 (79.4) | 96 (20.6) | <0.0001 |
| >3 to ≤6 | 97 (80.8) | 23 (19.2) | |
| >6 to ≤10 | 19 (73.1) | 7 (26.9) | |
| Breathing Problem | n=1323 | n=237 | |
| =0 | 504 (79.0) | 134 (21.0) | |
| >0 to ≤3 | 542 (90.2) | 59 (9.8) | <0.0001 |
| >3 to ≤6 | 219 (85.6) | 37 (14.5) | |
| >6 to ≤10 | 58 (89.2) | 7 (10.8) | |
| Fatigue | n=1311 | n=237 | |
| =0 | 261 (98.5) | 4 (1.5) | |
| >0 to ≤3 | 86 (76.1) | 27 (23.9) | <0.0001 |
| >3 to ≤6 | 314 (80.9) | 74 (19.1) | |
| >6 to ≤10 | 650 (83.1) | 132 (16.9) | |
| Pain | n=1324 | n=237 | |
| =0 | 469 (84.7) | 85 (15.3) | |
| >0 to ≤3 | 635 (85.7) | 106 (14.3) | 0.0290 |
| >3 to ≤6 | 195 (85.2) | 34 (14.9) | |
| >6 to ≤10 | 25 (67.6) | 12 (32.4) | |
| Vomiting | n=1320 | n=237 | |
| =0 | 1246 (85.5) | 212 (14.5) | |
| >0 to ≤3 | 54 (74.0) | 19 (26.0) | 0.0103 |
| >3 to ≤6 | 16 (84.2) | 3 (15.8) | |
| >6 to ≤10 | 4 (57.1) | 3 (42.9) | |

Note—frequency presented row wise.

## Limitations

This was a prospective cohort study and there is no usual care comparator, and the model was delivered in the Australian healthcare context which may limit

**Table 4** Univariate analyses of symptom distress score by symptom for mode of separation (admitted to hospital or palliative care unit vs died at home)

| Symptom score | OR | 95% CI | P value |
|---|---|---|---|
| Insomnia | 1.13 | 1.048 to 1.221 | 0.0015 |
| Appetite | 1.11 | 1.048 to 1.168 | 0.0003 |
| Nausea | 1.31 | 1.178 to 1.465 | <0.0001 |
| Bowel | 1.13 | 1.054 to 1.218 | 0.0007 |
| Breathing problem | 0.88 | 0.812 to 0.945 | 0.0006 |
| Fatigue | 1.09 | 1.041 to 1.144 | 0.0003 |
| Pain | 1.05 | 0.973 to 1.126 | 0.2175 |
| Vomiting | 1.21 | 1.063 to 1.377 | 0.0040 |

generalisability of this model elsewhere. The majority of the recipients had cancer.

In this study, only symptoms assessed by the SAS were included in the analysis; there is possibility that patients might have other prominent or bothersome symptoms which may be unique to a particular palliative diagnosis that could have resulted in separation from the programme.

As the data collection ceased at separation from the package, the ongoing or change in symptom distress until death for patients admitted to hospital was unknown, though it is likely that these patients died during that index admission.

### Recommendations

When designing home-based palliative care models aimed to support palliative care patients at home, the model needs to be tailored to particular needs, time points in the illness trajectory and preferences and evaluated to ensure ongoing capacity to refine the model to optimise patient outcomes. Further studies need to include more participants with non-cancer diagnoses. There is opportunity to further intensify support and symptom management when symptom distress is not improving or increasing, and fatigue warrants specific attention.

## CONCLUSION

This study findings suggest that not only does the PEACH model of care allow people with palliative diagnoses to

**Table 5** Multivariate analyses (with forward selection) of symptom distress score by symptom for mode of separation (admitted to hospital or palliative care unit vs died at home)

| Symptom score | OR | 95% CI | P value* |
|---|---|---|---|
| Nausea | 1.23 | 1.10 to 1.38 | 0.0003 |
| Breathing | **0.86** | **0.79 to 0.93** | 0.0001 |
| Fatigue | 1.08 | 1.03 to 1.14 | 0.0029 |
| Insomnia | 1.10 | 1.01 to 1.20 | 0.0226 |
| Bowel | 1.09 | 1.02 to 1.19 | 0.0187 |

Bold values signifies P<0.0001
*With the adjustment of preferred place of death and diagnostic category.

meet their preference to die at home, it achieves this while maintaining symptom control. Response to increase in particular symptoms may further optimise these models of care.

**Author affiliations**
¹South Western Sydney Local Health District, Liverpool, New South Wales, Australia
²Ingham Institute of Applied Medical Research, Liverpool, New South Wales, Australia
³Sydney Local Health District, Camperdown, New South Wales, Australia
⁴Illawarra Shoalhaven Local Health District, Wollongong, New South Wales, Australia
⁵Nepean Blue Mountains Local Health District, Nepean, New South Wales, Australia
⁶Clinical Innovation & Business Unit, South Western Sydney Local Health District, Liverpool, New South Wales, Australia

**Contributors**  MA, JSFC, WX and NM developed the idea and methodology of the study, provided governance and oversight and draft manuscript. JSFC, KJ and JH helped conduct the study, reviewed draft manuscripts. JSFC and KJ provided support and guidance of staff. MA, WX and NM analysed and interpreted the data. JL, GB and AO provided advice about study planning. JSFC, MA, WX and NM drafted the protocol and manuscript. All authors approved the final version of this manuscript. The authors accept full responsibility for the work and/or the conduct of the study, had access to the data, and controlled the decision to publish.

**Funding**  The authors have not declared a specific grant for this research from any funding agency in the public, commercial or not-for-profit sectors.

**Competing interests**  None declared.

**Patient and public involvement**  Patients and/or the public were not involved in the design, or conduct, or reporting, or dissemination plans of this research.

**Patient consent for publication**  Not applicable.

**Ethics approval**  Ethical approval was granted from South Western Sydney Local Health District Human Research Ethics Committee (LNR/15/LPOOL/591) and Silver Chain Human Research Ethics Committee (EC App 144).

**Provenance and peer review**  Not commissioned; externally peer reviewed.

**Data availability statement**  Data are available upon reasonable request. Data may be obtained from a third party and are not publicly available.

**ORCID iD**
Josephine Sau Fan Chow http://orcid.org/0000-0002-8911-6856

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
