## [Reviewer comments · BMJ Open]

ARTICLE DETAILS

TITLE (PROVISIONAL)	Longitudinal symptom profile of palliative care patients receiving a nurse-led end of life (PEACH) Program to support preference to die at home
AUTHORS	Agar, Meera; Xuan, Wei; Lee, Jessica; Barclay, Gregory; Oloffs, Alan; Jobburn, Kim; Harlum, Janeane; Maurya, Nutan; Chow, Josephine

VERSION 1 – REVIEW

REVIEWER	Zhuang, Qingyuan National Cancer Centre Singapore, Division of Medical Oncology
REVIEW RETURNED	22-Feb-2023

GENERAL COMMENTS	1) The PEACH package is described as for those who are terminally ill and wish to die at home. Yet, 11% of the cohort did not prefer to die at home. Was this an error in cohort identification? 2) Table 1 should specify the total numbers in the 3 separation groups for more clarity 3) What was the median duration of follow-up from recruitment to separation for the entire cohort and for each of the 3 separation groups? - this is all the more pertinent when we are studying symptom change comparing baseline and at separation. If the median duration of follow-up is long, then the limitation here is the assumption that symptoms do not fluctuate over time (which may not be an ideal assumption) 4) There are multiple statistical comparisons being done in Table 3 for the entire panel of symptoms. If correction was done (e.g. bonferroni), how many comparisons would still remain statistically significant? This needs to be addressed 5) Table 4 represents Table 3 in a categorical form for symptoms severity. Why not a chi-square test for comparison. The OR presented here is quite confusing 6) I am uncertain about the statistical relevance of doing forward selection of symptoms scores to build the multivariate logistic model. - why not adjust for relevant confounders for each symptom (e.g. ACP preferences, comorbidity, age, etc) to see whether it maintains significance as a predictor for hospital admission? - Will however defer to a formal statistical opinion
--

	7) What is the frequency that the ESAS was done. How much missing data is there for each symptom within the ESAS within the whole cohort. And if there was significant missingness, how was the missingness handled? 8) Agree with findings that high symptoms increase risk of rehospitalization. This was also true in a home-based palliative care cohort study where high symptom needs increased incidence rate ratios of acute healthcare utilization. https://doi.org/10.1186/s12916-022-02513-y 9) Limitations: Assumption that symptoms do not fluctuate but follow a linear gradient (the comparison here is baseline and separation). One possible way would be to conduct symptom trajectory analysis instead of comparing the difference between baseline and end.
--	---

VERSION 1 – AUTHOR RESPONSE

Reviewer 1:

1) The PEACH package is described as for those who are terminally ill and wish to die at home. Yet, 11% of the cohort did not prefer to die at home. Was this an error in cohort identification?

Response: We have modified in the manuscript which will clarify the question. Please refer to Methods section (Study Intervention), page 9.

'PEACH enables clients in their deteriorating and/or terminal phase to die at home or to stay at home as long as possible as per their wishes. Patients who were undecided about their preferred place of death but expressed an interest in attempting to stay home for end-of-life care or who wished to remain at home and transfer to hospital when death was imminent were also provided with a PEACH package. It was recognised for some clients there are a number of factors such as confidence in ability to die at home by either the patient and/or carer as well as cultural, religious, education or family demographics such as younger children in the home, impacted on preference for end-of-life care, i.e. undecided or transfer when death imminent. however, that should not in itself prohibit the client from receiving a package. Achieving end-of-life care preference was considered as one of the main objectives of the Program'.

2) Table 1 should specify the total numbers in the 3 separation groups for more clarity.

Response: As suggested, we have added relevant information to Table 1.

3) What was the median duration of follow-up from recruitment to separation for the entire cohort and for each of the 3 separation groups?

- this is all the more pertinent when we are studying symptom change comparing baseline and at separation. If the median duration of follow-up is long, then the limitation here is the assumption that symptoms do not fluctuate over time (which may not be an ideal assumption)

Response: We have added the further information in the manuscript, Methods section (Data collection and variables), page 10.

'The median duration for patients on the program was 6 days. All patients as per the model of care were in receipt of daily clinical visits, which included a daily symptom score assessment until separation from the program. At times some clients/carers declined the daily visit and hence SAS would not be recorded for that day'.

4) There are multiple statistical comparisons being done in Table 3 for the entire panel of symptoms. If correction was done (e.g. bonferroni), how many comparisons would still remain statistically significant? This needs to be addressed,

Response: We acknowledge the issue of multiple comparisons that the reviewer raised. The message presented in table 3 is (renamed as Table 2): while symptom scores collected at baseline were not associated with admitted to hospital, symptom scores measured in the end were associated. In terms of dealing with multiple domains of symptom and the possible correlations among them, we implemented a multivariable logistic regression analysis with forward variable selection procedure for the purpose to identify which symptom domains would be independently associated with admitted to hospital. Please refer to response to point 6 for further information.

5) Table 4 represents Table 3 in a categorical form for symptoms severity. Why not a chi-square test for comparison. The OR presented here is quite confusing

Response: As suggested we have revised Table 4 (renamed as Table 3), Chi square test now used and p-value presented in Table 3; OR and CI removed.

6) I am uncertain about the statistical relevance of doing forward selection of symptoms scores to build the multivariate logistic model.

- why not adjust for relevant confounders for each symptom (e.g. ACP preferences, comorbidity, age, etc) to see whether it maintains significance as a predictor for hospital admission?
- Will however defer to a formal statistical opinion

Response: Logistic regression with forward selection was used to identify the independent symptom scores that were associated with admitted to hospital/palliative care unit. The univariate analysis presented in Table 3 only provide an indication on the possible associations. Given the symptom scores might be correlated to each other and may not independently “predict” the dependent variable, implementing multivariable logistic regression with forward variable selection can tell us the subset of the symptom score measures which were indecently associated to the dependent variable.

7) What is the frequency that the ESAS was done. How much missing data is there for each symptom within the ESAS within the whole cohort. And if there was significant missingness, how was the missingness handled?

Response: Among the PEACH cohort (N=1327+237+190=1754, Table 3), the majority of the patients had final symptom score information collected. Among the 8 symptom score domains, 7 final symptom scores were collected for at least 97.6% (=1712/1754) of the sample, with the exception of Appetite symptom where N=1604 (91.5%) who had final symptom score collected. We acknowledge the data were incorrectly presented in table 3 of the previous version. The baseline symptom scores in Table 3 were presented with N=1586+303+214. This sample includes additional patients whose baseline measurements were stored in the database but does not belong to the analysed PEACH cohort. This Table is now revised and the correct baseline data of the PEACH cohort (N=1754) are presented in this Table.

8) Agree with findings that high symptoms increase risk of rehospitalization. This was also true in a home-based palliative care cohort study where high symptom needs increased incidence rate ratios of acute healthcare utilization. <https://doi.org/10.1186/s12916-022-02513-y>

Response: We have added relevant reference in the Discussion section, page 24

9) Limitations:

Assumption that symptoms do not fluctuate but follow a linear gradient (the comparison here is baseline and separation). One possible way would be to conduct symptom trajectory analysis instead of comparing the difference between baseline and end.

Response: We agree with the reviewer on this view. The symptom trajectory analysis will be able to investigate the complex pathway including non-linear gradient on change in symptoms that might be

associated with admitted to hospital/palliative care unit. We acknowledge this limitation and may implement future research on this pathway.

VERSION 2 – REVIEW

REVIEWER	Zhuang, Qingyuan National Cancer Centre Singapore, Division of Medical Oncology
REVIEW RETURNED	06-Oct-2023

GENERAL COMMENTS	Thank you for getting me to review this manuscript. Previous comments have been addressed by the author. However, there seems to be a formatting issue with Table 3? Column of "Died at Home" percentage is wrong throughout entire column e.g. 942 (87.1) and 266 (79.2) for insomnia =0 and insomnia <=3 respectively? Because the previous PEACH study identified preference for home death to be significantly associated with a successful home death, this seems to be a significant variable that should be considered and adjusted for within the multivariate analysis? I wonder whether after accounting for home death preferences, are all higher symptom distress scores still significantly associated with admission to hospital/pcu?
--

REVIEWER	Millares Martin, Pablo Whitehall Surgery
REVIEW RETURNED	15-Oct-2023

GENERAL COMMENTS	A very interesting article in the matter of EOL care, although I was wondering about a few little things:  -A simple grammar error in page 6, second paragraph, as it should be cared not care. -How much contact was given by the PEACH assistant in nursing through the day? -What was the minimum and maximum duration of the program? -Were any patient discharged from the program and later rejoined? -Why symptom assessment score (SAS) used instead of other like Edmonton symptom assessment system -Why using a reference for the SAS from 2021 rather than one from before the start of the study, like Aoun SM, Monterosso L, Kristjanson LJ, McConigley R. Measuring symptom distress in palliative care: psychometric properties of the Symptom Assessment Scale (SAS). J Palliat Med. 2011 Mar;14(3):315-21. doi: 10.1089/jpm.2010.0412. -How long was PEACH used in each of the three modes of separation? Was it statistically different? -How many patients failed to have their preferred place of death at home? One consider the manuscript could benefit from getting a bit deeper into the benefits of care at home looking at the above questions.
---

VERSION 2 – AUTHOR RESPONSE

Reviewer 1:

1) There seems to be a formatting issue with Table 3? Column of "Died at Home" percentage is wrong throughout entire column e.g. 942 (87.1) and 266 (79.2) for insomnia =0 and insomnia <=3 respectively?

Because the previous PEACH study identified preference for home death to be significantly associated with a successful home death, this seems to be a significant variable that should be considered and adjusted for within the multivariate analysis? I wonder whether after accounting for home death preferences, are all higher symptom distress scores still significantly associated with admission to hospital/pcu??

Response: The frequency is presented row wise in Table 3.

Table 5, representing multivariate analysis, has been updated with adjustment of preferred place of death and diagnostic category.

Reviewer 2:

1) A simple grammar error in page 6, second paragraph, as it should be cared not care.

Response: As suggested, the error has been corrected.

2) How much contact was given by the PEACH assistant in nursing through the day?

Response: Added details in the manuscript, Methods section (Study Intervention)

'The PEACH Assistant in Nursing provided a personal care visit or respite for up to 1 hour per day, 7 days per week until the client separated from the Program.'

3) What was the minimum and maximum duration of the program?

Response: Added in the manuscript, Results section.

'The duration of stay on the program for clients ranged from <24 hrs to 2 months.'

4) Were any patient discharged from the program and later rejoined?

Response: Added in the manuscript, Results section.

'Patients who were discharged and came back onto the program at a later date, they were counted as another patient encounter. Thirty-three clients who were discharged from the Program re-joined later.'

5) Why symptom assessment score (SAS) used instead of other like Edmonton symptom assessment system?

Response: The SAS score is utilised in the Palliative Care Outcomes Collaboration (PCOC) data set in Australia. It is a validated tool.

6) Why using a reference for the SAS from 2021 rather than one from before the start of the study, like Aoun SM, Monterosso L, Kristjanson LJ, McConigley R. Measuring symptom distress in palliative care: psychometric properties of the Symptom Assessment Scale (SAS). J Palliat Med. 2011 Mar;14(3):315-21. doi: 10.1089/jpm.2010.0412

Response: Reference in the manuscript has been updated.

7) How long was PEACH used in each of the three modes of separation? Was it statistically different?

Response: The median duration on the program was 6 days. The duration of the stay was same irrespective of the mode of separation.

8) How many patients failed to have their preferred place of death at home?

Response: Updated in the manuscript, Results section, page 12.

'Almost, 78.7% of the clients who indicated they preferred to die at home met their wish, 55.6 % of clients who were unsure/undecided of preferred place of death on referral to the Program died at home.'